# Printed Versus Electronic Texts in Inclusive Environments: Comparison Research on the Reading Comprehension Skills and Vocabulary Acquisition of Special Needs Students

**Özlem Dağlı Gökbulut [1] and Ahmet Güneyli [2],***  

[1]   Department of Special Education, Cyprus International University, Nicosia 99258, Northern Cyprus; odagli@ciu.edu.tr

[2]   Department of Turkish Language Education, Faculty of Education, European University of Lefke, Mersin 10 Turkey, 99728, Northern Cyprus

*   Correspondence: aguneyli@eul.edu.tr; Tel.: +90-542-855-0047

**Abstract:** In this research, an effort is made to compare the effectiveness of reading texts presented through electronic books in a computer environment and regular (printed) texts, in terms of the development of the reading comprehension and vocabulary acquisition skills of students with special needs within inclusive educational environments. The research was designed with 'Adaptive Alternating Applications', and the study group of the research was formed by using the 'purposive sampling' method. As a requirement of the research design used, two special needs students were studied. According to the results of the study, the vocabulary acquisition levels of both students produced more effective results in presentations made with electronic texts, and in addition, electronic texts were found to be more effective in improving reading comprehension skills than printed texts.

**Keywords:** special education; inclusion; reading comprehension skills; electronic texts; printed texts

## 1. Introduction

The inclusive education of students with special needs, beginning in the pre-school period with their normal peers, and based on the principle of the equality of opportunity creates satisfactory results: The academic foundations of these students become stronger, and they integrate better with their peers and with society at large. After reaching basic literacy goals in the first years of primary school, all students with and without special needs in inclusive education environments shift to the more complex and higher dimensions of reading comprehension skills in the early years of primary school [1]. Through the inclusive activities carried out at primary level, students are expected to achieve success in social development and basic academic skills, provided by courses such as Turkish Language, Mathematics and Life Sciences, and to achieve the objectives determined by the curriculum [2]. In the literature, it is reported that due to various neurological reasons (mental disabilities, distraction and loss of hearing or sight) or, despite having adequate mental and neurological development and favorable learning environments, due to learning difficulties, where the source is not clearly identified (dyslexia), and also due to inadequate environmental conditions, problems can be experienced throughout the entire school life in almost all academic skills [3–10].

Reading comprehension skills, which have a fundamental relationship with all courses, are the primary academic skills that students with special needs frequently find problematic in inclusive education environments [7,11].

In classes where courses are taught with traditional teaching methods and oral presentations on the basis of age and class-related programs, it is often difficult for students with special needs to attend classes and to easily understand the information transmitted. This situation can manifest itself in the form of reading errors in students with special needs, the inability to enjoy reading, delays/retardation in vocabulary acquisition, and the inability to understand reading texts fully and accurately, all of which can result in a decline in general school achievement [7,11–19].

Using assistive technologies and various instructional adaptations on the basis of individual differences between students by classroom teachers, planning all the courses related to reading comprehension skills is considered as an important and effective factor in increasing the success of all students in the classroom, particularly students with special needs who have problems in reading comprehension skills [19–28]. According to the United Nations Educational, Scientific and Cultural Organization (UNESCO) Global Report [29], in classrooms where assistive technology is successfully applied, the physical environment and facilities are more accessible, the techniques used in the context of learning and teaching better meet the needs of students, and a more inclusive educational environment is created. Today, the use of technology in educational environments to support the academic development of students is becoming more popular in most parts of the world [30]. Hoeckner [31] emphasizes that most of the innovative changes that have been developing have had a strong impact on inclusive education practices and legal arrangements in the infrastructure for nearly 20 years.

Assistive technologies that encompass different types of technology, such as devices, tools and computer hardware or software, enrich the learning environment and create more favorable learning opportunities. In this regard, assistive technologies enable students with special needs in inclusive environments to be productive in areas of the curriculum where they have difficulty, and such technologies are highly effective in generalizing the learned skills [25,27,32–39]. Eliçin [40] states that in classroom settings, determining the most appropriate technologies for all students, with and without special needs, with different learning characteristics and needs, is the most important step that teachers should take in using technology.

According to Strouse and Ganea [41], the reading activity, which is one of the important academic aspects of school life, and which was previously only possible through printed materials, has started to change today. This change has begun with the introduction of technologies such as touch screens, computers and smartphones. In addition to reading through printed texts, electronic reading (e-reading) rates have also increased. Korat and Shamir [42] report that electronic reading (story) books are one such type of software, referring to the fact that school-age children can access unlimited software with the help of technological resources at home and at school. Korat and Shamir [42] also emphasize that the lively and engaging multimedia effects of such storybooks have an impact on supporting literacy and language development in young children.

As it is known, the reading comprehension skills and vocabulary acquisition levels of students with different educational abilities and individual discrepancies may vary, depending on several factors. One of these factors is the presentation style of the given text (printed or electronic). It is stated that students' level of understanding of texts given in printed and electronic media may vary, even though their characteristics and contents are the same [43]. In terms of literature surveys, it has been stated that high scores have been achieved in reading comprehension skills in a number of studies conducted using printed texts [44,45]. On the other hand, it has been observed that higher results have been obtained in the studies conducted through electronic texts [42,46] compared to those based upon printed texts. The reason for this is that the visual, animation-like features in the structure of electronic texts that appeal to various senses help the readers to keep/perceive the message [43,46].

## 1.1. The Objective of the Study

In this research, an effort is made to compare the effectiveness of reading texts presented through electronic books in a computer environment, and regular (printed) texts, in the development of the

reading comprehension and vocabulary acquisition skills of students with special needs in inclusive education environments. For this purpose, answers to the following questions were sought in the research:

In regard to the students with reading comprehension skills difficulties in inclusive education settings,

i　　Are there any differences in the total correct answers to the vocabulary acquisition questions between the printed and electronic texts offered to the students?

ii　　Are there any differences in the total correct answers in terms of the development of reading comprehension skills questions between the printed and electronic texts offered to the students?

iii　　According to the social validity results of the students, are there any differences in terms of the vocabulary acquisition and reading comprehension skills concerning the printed and electronic texts?

## *1.2. Importance of the Study*

It has been found that the reading skills of students with special needs within inclusive education environments can be affected by various factors, and this effect often manifests in terms of difficulties in reading and literal skills performance. It is stated in the literature that these difficulties decelerate academic development, or cause students to lag behind the level expected, compared to their peers [2,15–17]. In both general and special field studies, where the effects of some adaptations in the educational environments on the development of academic achievement have been examined [20,22,23,26], the use of assistive technologies in the classroom has been reported to support the achievement of effective results in the development of literal skills [36,42,46,47]. The results obtained from this research are thought to be important in terms of revealing the effect of assistive technology, which is an effective alternative in increasing the academic achievement of students with special needs in classroom environments where traditional teaching methods are applied through printed texts and oral expressions.

## 2. Materials and Methods

In this section, the materials and methods of this study are explained.

### *2.1. Subjects*

The study group of the research was formed by using the 'purposive sampling' method, and as a requirement of the research design used, two special needs students were studied. Purposeful sampling is a method that allows an in-depth study of situations believed to be rich in information [48]. Accordingly, the study was conducted with the participation of two male students aged 7 and 8 who had difficulty in literal skills and who were attending the second grade of a primary school affiliated to the Ministry of National Education in Northern Cyprus. The students had been diagnosed with learning disabilities by the Guidance and Research Center of the Ministry of National Education, and had difficulties in terms of reading comprehension. These students also received special education support from the resource room at their school.

The following criteria were set for the students who participated on a voluntary basis:

1.　A word recognition level of at least 60% of words in a text
2.　The ability to read a minimum of 30 words per minute
3.　Having read at least 15 correct words
4.　Providing three or less correct answers to the literal skills questions for the text they have read
5.　The number of new words acquired in the text being read is three or less.

Two students, and two others who fulfilled the prerequisites, were identified as subjects and substitutes. In the study, in accordance with ethical rules and participant confidentiality, the real names

of the subjects were not used, and the students were given pseudonyms. The first language of both of the students with special needs is Turkish.

The characteristics of the subjects are described in the following table (Table 1):

**Table 1.** Characteristics of the Subjects

| Name of the Subject | Age | Number of Words Read in 1 min | Number of Words Accurately Read in 1 min | Number of Correct Answers Given to Literal Skills Questions | Number of Correct Answers Given to Vocabulary Questions |
| --- | --- | --- | --- | --- | --- |
| (1.Subject) Hikmet (pseudonym) | 7 | 32 | 15 | 6 | 2 |
| (2.Subject) Arda (pseudonym) | 8 | 31 | 17 | 5 | 2 |

When the table is examined, it is seen that the first subject, Hikmet, was able to read a total of 32 words per minute, 15 of which were read accurately. It is seen that the new vocabulary acquisition level of Hikmet in reading texts is Level 2, and that he gave six correct answers to the literal skills questions on the text. On the other hand, it can be seen that the second subject, Arda, could read 31 words per minute, and expressed 17 of these words accurately. According to the table, the new vocabulary acquisition level of Arda in reading texts is Level 2, and he gave five correct answers to the literal skills questions on the text.

*2.2. Application Setting*

Applications were completed in a one-to-one environment in the resource room of the school at which the students were registered. Accordingly, the eight square-meter resource room had a standard student desk, two chairs, an educational material locker, a whiteboard, a laptop, a camera for recording the process, and a material box. Prior to commencing the applications, a camera was position in the classroom in a location where it would be installed during the main applications, and it was ensured that the students became accustomed to the presence of the camera in the classroom.

On the days of application, the physical properties of the room (heat, light, sound, noise, readiness of the equipment and battery and brightness characteristics of the laptop and camera) were checked before each session. These media arrangements were prepared in the same order prior to each session. A sketch reflecting the physical properties of the resource room in which the applications were carried out is illustrated in Figure 1.

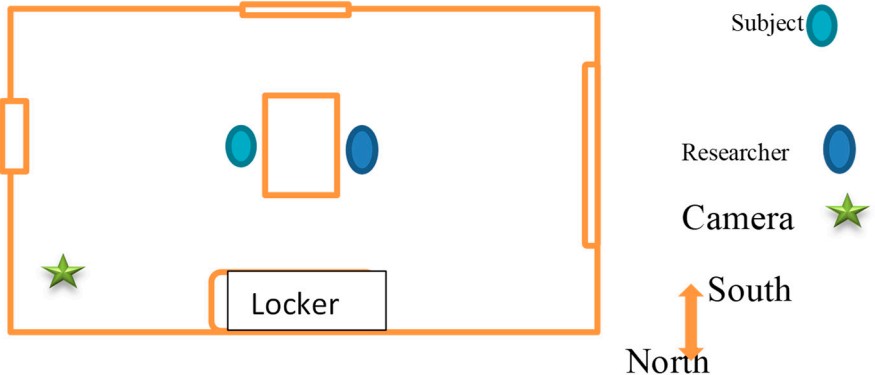

**Figure 1.** Physical properties of the resource room where the applications were executed.

As shown in Figure 1, illumination was provided from the North (parallel to the classroom door) through a window of medium size, and when entering the entrance door of the resource room in which the applications were implemented. The room contained a student desk with standard dimensions located at a distance of three meters from the window and in line with the door. Two student chairs were placed in front of and behind the desk, respectively. A large whiteboard was also mounted upon the wall behind the desk. A three-door material locker was positioned on the right side of the whiteboard and parallel to the student desk. The camera, which was used to record data during the applications, was positioned in the left diagonal of the cupboard and to the left of the entrance door, in order to record the subject student from behind, and the researcher from the front. During the experimental process, the subject was always seated with his back to the entrance door and the camera facing the blackboard, while the researcher sat directly opposite him.

*2.3. Tools and Supplies*

When comparing the effect of the two independent variables (printed and electronic texts), the 'Let's Learn with Fun' reading set, consisting of three books containing printed-electronic texts and CDs of the book contents, were used. The reason for preferring this set during applications was that all of the printed and electronic texts used as a part of this set are identical in terms of their layout and appearance. Maintaining the uniformity of the texts and the page structures enabled the texts in all stories to be presented in the same size and in a uniform page layout. The only difference between the texts is that one of them is printed and the other is electronic.

The characteristics of the texts used during the application were as follows: Stories were selected by analyzing the difficulty level and the number of words. The texts consisted of structure, décor, introduction, development and conclusion sections, and each section included a plot containing the initiator event, reaction to the initiator event, situations developed after the reaction, and the final reaction/behavior presented by the main character depending on these situations. The length of these texts was limited to 150–550 words on average. The narration was performed by a third party, and the word structures were used with simple word sequences, and avoiding metaphorical discourses.

The session schedule is presented in Table 2.

**Table 2.** Weekly routine work and session schedule.

| 1. Week | Monday | Tuesday |
|---|---|---|
| Title of the printed text | Two surprises at a time | Pengu learns to swim |
| Title of the electronic text | Ceren's season garden | Little turtle plays hide-and-seek |
| **2. Week** | **Monday** | **Tuesday** |
| Title of the printed text | Little car looking for a friend | Pengu and the baby whale |
| Title of the electronic text | Pengu's birthday | Cute little bird |
| **3. Week** | **Monday** | **Tuesday** |
| Title of the printed text | Pengu's dream | Small raindrop |
| Title of the electronic text | Sarman looking for the way home | Pengu and the sea lion |

Baş and Yıldız's [49] study on the readability of Turkish texts was used to balance the readability of the printed and electronic texts. The Mean Length of Utterances (MLU) and Average Sentences Length (ASL) values used for the second-year students (7–8 years old) were taken as references. Accordingly, taking into account the second grade level of elementary education, attention was paid to ensure that the MLU varied between 2.00 and 3.00, and the ASL values varied between 4.00 and 8.00. It was noted that the MLU and ASL values of the texts (electronic-printed) used alternatively on the same day, as stated in Table 2, were similar. In addition, the total number of words in the text, the number of sentences, and the number of the unknown words, were nearly equal. The texts used were from the same publisher, and the authors were also the same. Finally, the opinions of two Turkish language

teaching experts on the difficulty of the texts were also consulted. Four pilot-study sessions were conducted with an 8-year-old student who had difficulty in reading comprehension, although he had not previously been diagnosed as a child with special needs.

In the following pilot application, the developments in the process were carefully monitored, and two texts, which were determined to have the potential to affect the main application due to the difficulty of the texts, were excluded from the program.

*2.4. Data Collection Tools*

Separate data collection tools were used to determine reading speed, word recognition percentage, total and correct word numbers read in one minute, which are among the prerequisite skills mentioned in the selection of subjects, as well as to identify literal skills levels. The "Reading Evaluation Tool Form", used to determine whether the subjects met the prerequisite skills, was developed by Akyol et al. [50]. It is a form based on a standard text suitable for the relevant age group, through which the reading speed per minute, prosody, total reading points and the number of accurately read words, can be indicated on the paper for each student.

Reading comprehension is a broad concept regarding how text is processed by a person. It involves literal skills and inferential skills. In this study, the aim is to measure only the literal skills of the students, and therefore the objective is to only address the responses to "wh questions" as one aspect of comprehension. The 'Reading Skills Follow-up Chart', on which the performance in literal skills and vocabulary acquisition levels are recorded, is a chart that can be used to mark the number of right and wrong answers given to the questions about the text and the items evaluating the word acquisition. In this chart, the words 'true' and 'false' are expressed for each vocabulary acquisition and literal skills goal expected from the students. At the end of each session, students marked the appropriate items during the evaluation of their outputs, and the correct number of answers were determined.

The questions on the vocabulary acquisition and literal skills level related to the texts used in the research were prepared by the researchers by examining the printed-electronic texts. In the vocabulary acquisition questions, new words in the text were chosen, and their meanings were explained. The questions that measure the level of literal skills consist of 10 open-ended and short-answer questions about the text (asking who, where and when the event took place, what happened in the text, and what happened at the end). (See Attachment 1).

At the end of the study, social validity data were collected from the subjects. Social validity data were obtained by using a semi-structured form prepared by the researchers. It was ensured that the short-answer questions used in the semi-structured opinion form were easy to understand, and nine questions were used to facilitate a general assessment of the overall process.

*2.5. Research Model*

The research was designed with the 'Adapted Alternating Treatment' model, which is one of the single subject research models. This research model is a comparison of the effectiveness of two or more independent variables on two or more irreversible dependent variables. In this model, there are functionally similar but independent variables with equal difficulty levels for each independent variable. According to the model, applications of independent variables should be transformed rapidly [51].

2.5.1. Dependent and Independent Variables

The dependent variables of the study were determined as the students' literal skills and vocabulary acquisition levels in a Turkish language course. "Wh questions" are used after reading texts, so reading comprehension skills are limited to literal skills in the study. The independent variables were determined as printed (plain) texts and electronic texts appropriate to the level of the students. In addition, in order to control the effect of the factors other than the independent variable on the experimental process, equally challenging and functionally similar but independent texts were selected.

In the study, experimental control was established by providing a rapid transformation of independent variables on dependent variables by giving an interval of at least one hour between applications made with both text types on the same day.

The data related to the dependent variables measured in the study (literal skills and vocabulary acquisition) were obtained using the Reading Skills Follow-up Chart explained in the Data Collection Tools section. This checklist includes a list of new words that are targeted for acquisition, and mostly "wh" questions were used to measure text comprehension. These checklists created for each text were evaluated after the student had answered all the questions and returned the paper to the researcher. True answers and false answers are marked by (x) in the checklist, and the total score was calculated by giving '1' for true answers and '0' for false and missing answers. The total scores of the dependent variables indicated in the tables were determined in the same way. Additional information about the data collection tool is provided in the Data Collection Tools section.

### 2.5.2. Research Process

Before the study data were collected, the aim of the study and the process to be followed were explained to the students, parents and school management who had agreed to participate in the study, and their participation in the study was provided on a voluntary basis. Prior to starting the experimental process, the students' literal skills and vocabulary acquisition beginner level data were obtained by repeated measurements in three different beginner level sessions.

### 2.6. Application Process and Collection of Data

The application took place over a period of three weeks between March and April in the 2018-2019 academic year. Every subject was treated for a maximum of two days a week and two sessions a day. Each session was limited to 20 min. A one-hour break was given between the two sessions during the day, and in the course of these breaks, suitable arrangements were made in the program so that the students could engage in other classes/activities.

A total of 12 sessions involving 240 min of the application were performed with two subjects, with a total of 12 different texts (six printed, six electronic). The printed (plain) texts and electronic books used in this study were applied to each student on a rotational basis by drawing lots in advance. Before starting the applications, it was checked that the automatic voice reading features of the texts presented in each session electronically were turned off, and an environment was created that required the subjects to read the texts themselves.

During the experimental process, a researcher and a subject were present in the classroom. The subject and the researcher were seated opposite each other, while the student followed the steps stated below, and carried out the applications. Accordingly, the researcher introduced the printed/electronic text to be used before the application, and asked the student to read the text carefully. After the student had read and finished the story completely, the printed/electronic text was taken. The student was then asked to answer the questions that were printed on a worksheet consisting of questions assessing his vocabulary acquisition and literal skills. During the assessment, the researcher did not intervene or assist the child in any way. Following the completion of the questionnaire, the subject was thanked, and then allowed to return to his classroom.

After each application, the answers given to the measurement and evaluation tools, including vocabulary acquisition questions and literal skills questions about the texts, were evaluated, and further assessments were made on the correct number of answers. The experimental process was continued until the subjects (in the applications where both independent variables were presented) achieved 9/10 (answering nine out of ten questions) to text-related questions and 4/5 (explaining four out of five words) for the vocabulary acquisition measuring questions. In order to determine whether there was a difference in the effectiveness of the independent variables, both independent variables were presented to the subjects with the same instructions and the same evaluation process without any change during the applications, and the sessions were recorded via video camera to prevent any data loss.

The texts given in Table 2 were used in the implementation process carried out with both students. However, in the course of the program, if a student was first given electronic then printed text, the order was changed in the other step, whereby the electronic text was given first followed by the printed text. In addition, if the first application was made with one student (for example Arda) during the day, the same student was included in the practice on the second day. In this way, efforts were made to limit the effect of non-research factors, such as the time of day and the order of text presentation on the experimental process.

## 2.7. Analysis of the Data

In the studies carried out with an adaptive alternating applications pattern, the collected data were analyzed graphically. In the graph, the data of the dependent variable is recorded on the $y$ axis, and the data on the independent variable is recorded on the $x$ axis. The horizontal axis ($x$) corresponds to different time units, such as application sessions, class hours, days and weeks. The percentage or number of behavior occurrences corresponding to each session on this axis is plotted by marking the value on the $y$ axis. Interpretation is made by comparing the slope or level obtained as a result of treatment [50].

In this study, treatment curves and the initial levels of the students were taken into consideration in order to determine whether the students could achieve their goals. It was concluded that if the distance from the starting level curves and the horizontal $x$ axis is applied, the teaching that was administered leads to the students reaching their goals. In order to determine which of the two teaching methods is more effective, the slope path and direction of the students' level of skill acquisition during both teaching methods were compared. The arrangement to which the higher curve belongs was found to be more effective. In addition, it was concluded that if the slope path and direction of both teaching arrangements do not differ, they are similar in terms of effectiveness. Accordingly, printed (plain) and electronic texts were compared in terms of their slope path and direction in curves showing students' literal skills and recalling levels.

Validity-Reliability Analysis

Application reliability and inter-observer reliability data were also collected for the application sessions held during the research. Accordingly, as a result of the study's application reliability, which was evaluated with the formula "observed practitioner behavior/planned practitioner behavior x 100" [52], the applications carried out during the study were found to be 100% reliable. Before collecting this reliability data, the researchers introduced the 'Reading Skills Follow-up Chart' to the observers, and explained in detail what points they should pay attention to in the video camera recordings. Two independent observers evaluated the true answers (or false or missing answers) of the students' literal skills and vocabulary acquisition separately, and the researchers examined the consistency between these observers. Inter-observer reliability data of the study were obtained by a doctor of special education and an associate professor in the field of Turkish language teaching, viewing 40% of the video records. After determining the total correct answers by means of the checklist given to them, a calculation was made based on the 'consensus/consensus + disagreement × 100' formula [51] and the inter-observer reliability analysis was determined to be 100%.

## 3. Results

### 3.1. Comparison of the Effectiveness of Printed Texts and Electronic Texts in Terms of Vocabulary Acquisition Achievement Levels

Effectiveness data were obtained to determine whether there were any differences between the printed and electronic texts presented to the students with literal skill difficulties in the assessment of vocabulary acquisition achievement levels.

These data were acquired by collecting and calculating the percentages of true and false answers given by the subjects at the end of each session to the printed and electronic texts. Accordingly, the findings obtained from the subjects are indicated in the following tables (Tables 3 and 4):

**Table 3.** Comparison of the first subject Hikmet's vocabulary acquisition (Number of true answers given to the questions) level in terms of printed and electronic texts.

| Session Number | Printed Texts | Electronic Texts |
| --- | --- | --- |
| 1 | 0 | 3 |
| 2 | 2 | 4 |
| 3 | 2 | 5 |
| 4 | 3 | 4 |
| 5 | 4 | 5 |
| 6 | 5 | 5 |

As indicated in Table 3, in the sessions conducted with Hikmet using printed texts, he gave 0, 2, 2, 3, 4 and 5 true answers in terms of vocabulary acquisition in the first to sixth sessions, respectively. On the other hand, Hikmet's vocabulary acquisition performance in applications with electronic texts was determined as follows: Hikmet provided 3, 4, and 5 true answers in the first to third sessions of applications using electronic texts. In the 4th section, he regressed to four true answers, while he increased to five true answers again in the fifth and sixth sessions.

The data presented in the table above can be interpreted to mean that Hikmet's level of vocabulary acquisition in the printed text sessions started with zero words, and began to increase in the fourth session, ultimately reaching 100% in the fifth session. This situation suggests that Hikmet's speed of vocabulary acquisition with printed texts was progressing slowly. On the other hand, the fact that Hikmet's vocabulary acquisition level started at a high level after the first of the sessions applied with the electronic texts, then reached 100% in the third session and maintained this level until the last session, suggests that Hikmet achieved success in applications with electronic texts. In the most general interpretation, the differences in the level of vocabulary acquisition that Hikmet achieved with both text presentations suggest that Hikmet acquired vocabulary faster through the electronic texts.

As indicated in Table 4, in the sessions conducted with Arda using printed texts, he gave 2, 3, 2, 3, 3 and 3 true answers in regard to vocabulary acquisition in the first to sixth sessions, respectively. On the other hand, Arda's vocabulary acquisition performance in applications with electronic texts was determined as follows: Arda recorded 2, 3, 3, 4, 5 and 5 true answers in the first to sixth sessions applied with electronic texts.

**Table 4.** Comparison of the vocabulary acquisition skills (number of true answers given to the questions) of the second subject Arda in terms of printed and electronic texts.

| Session Number | Printed Texts | Electronic Texts |
| --- | --- | --- |
| 1 | 2 | 2 |
| 2 | 3 | 3 |
| 3 | 2 | 5 |
| 4 | 3 | 4 |
| 5 | 3 | 5 |
| 6 | 3 | 5 |

In the interpretation of the data in the table presented above, the fact that the level of vocabulary acquisition in the sessions with printed texts started with two words and increased to three in the following sessions, which was maintained until all sessions had been completed, suggests that Arda's level of vocabulary acquisition progressed very slowly. On the other hand, when the scores obtained in the electronic texts are examined, it is observed that Arda had a progressive course in the first sessions, as in the printed texts.

It can be seen that after reaching a level of 100% in the third session, Arda regressed to four correct response levels in the next session, and then succeeded to reach 100% again in the fifth and sixth sessions. This suggests that Arda was more successful, despite his varying performance in applications with the electronic texts. In general terms, it can be suggested that Arda's levels of vocabulary acquisition through both text presentations was progressive. However, it can be interpreted that Arda performed faster, but also exhibited more variable performance, especially with the electronic texts.

Figures 2 and 3 show the vocabulary acquisition changes of the students.

For vocabulary acquisition, the lowest score that students can achieve is 0, and the highest score is 5.

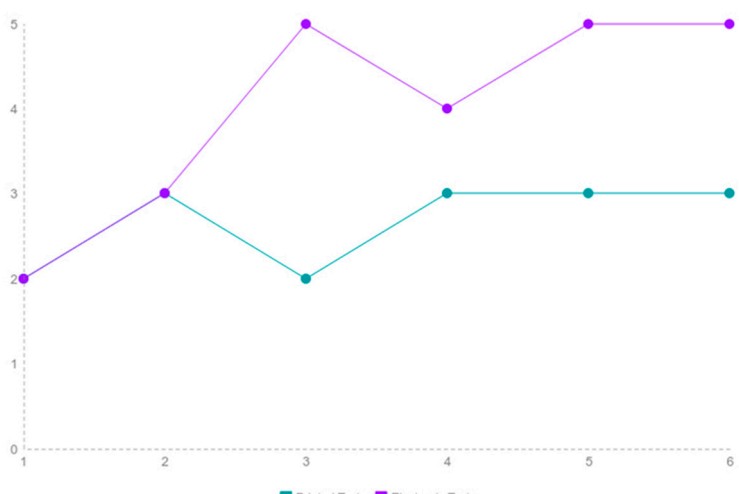

**Figure 2.** Vocabulary acquisition of Arda.

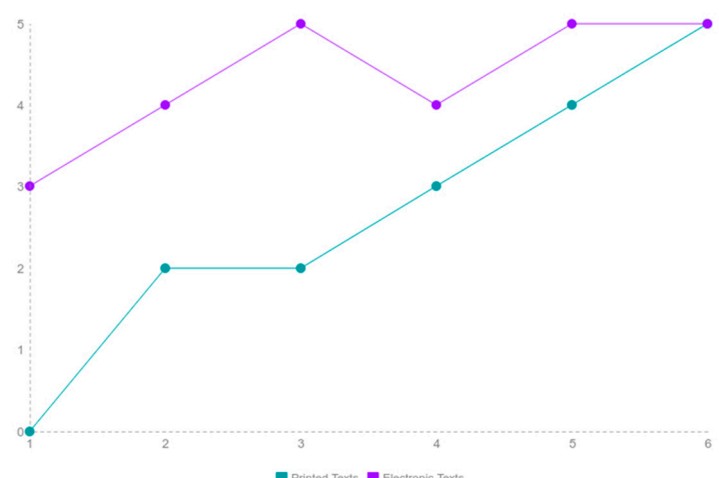

**Figure 3.** Vocabulary acquisition of Hikmet.

*3.2. Comparison of the Effectiveness of Printed and Electronic Texts in Terms of Literal Skills*

In assessing the development of the literal skills of students with reading comprehension difficulties, the effectiveness data for determining whether there is any difference between the printed texts and electronic texts presented to students, were obtained by means of the collection and calculation of the percentages of true and false answers given to "wh questions" prepared at the end of each session, using printed and electronic texts. Accordingly, the findings obtained from the subjects are indicated in the following tables (Tables 5 and 6):

**Table 5.** Comparison of literal skills of the first subject, Hikmet, in terms of printed and electronic texts.

| Session Number | Printed Texts | Electronic Texts |
|:---:|:---:|:---:|
| 1 | 6 | 6 |
| 2 | 6 | 8 |
| 3 | 7 | 8 |
| 4 | 6 | 10 |
| 5 | 8 | 10 |
| 6 | 8 | 10 |

As indicated in Table 5, the first subject Hikmet gave 6, 6, 7, 6, 8 and 8 true answers in the first to sixth sessions performed with printed texts, respectively. On the other hand, it can be seen that at the end of the sessions performed with electronic texts, Hikmet gave 6, 8, 8, 10, 10 and 10 true answers in the first to sixth sessions, respectively. Based on the obtained findings, it can be said that Hikmet gave more true answers to the literal skills questions in the sessions performed with electronic texts and that these electronic texts were more effective in developing Hikmet's performance.

**Table 6.** Comparison of literal skills of the second subject, Arda, in terms of printed and electronic texts.

| Session Number | Printed Texts | Electronic Texts |
|:---:|:---:|:---:|
| 1 | 5 | 6 |
| 2 | 6 | 8 |
| 3 | 7 | 8 |
| 4 | 6 | 9 |
| 5 | 6 | 10 |
| 6 | 7 | 10 |

As can be seen in Table 6, the second subject, Arda, gave five true answers at the end of the first session, six true answers at the end of the second session, seven true answers at the end of the third session, and the number of true answers was reduced to six in the fourth and fifth sessions. The subject again gave an increased number of seven true answers in the 6th session. On the other hand, at the end of sessions performed with electronic texts, it is seen that Arda reached six true answers at the end of the first session and eight true answers at the end of the second and third sessions. He reached nine true answers at the end of the fourth session. In the fifth session, it is seen that he achieved 10 true answers, corresponding to an accuracy of 100%, and in the sixth session, he maintained the same level of 10 true answers. In general, it can be concluded that the number of correct answers given to the literal skills questions in electronic sessions was higher than the reading comprehension studies conducted with printed (plain) texts, and that electronic texts were more effective in improving Arda's performance.

Figures 4 and 5 show the literal skills changes of the students.

The lowest score that students can achieve for literal skills is 0, and the highest score is 10.

In the tables and graphs above, the students' literal skills and vocabulary acquisition performances are tabulated and presented in electronic and printed texts in a comparative manner, and are thus presented in Tables 7 and 8.

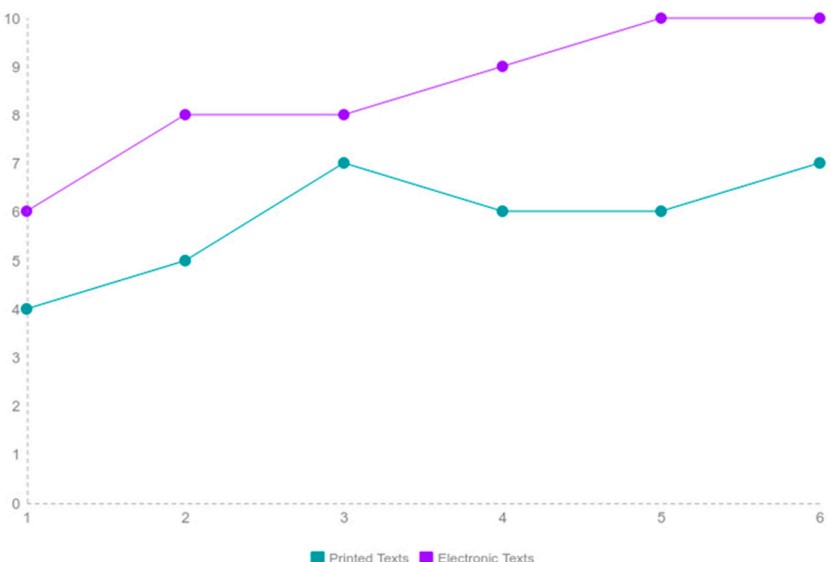

**Figure 4.** Literal skills of Arda.

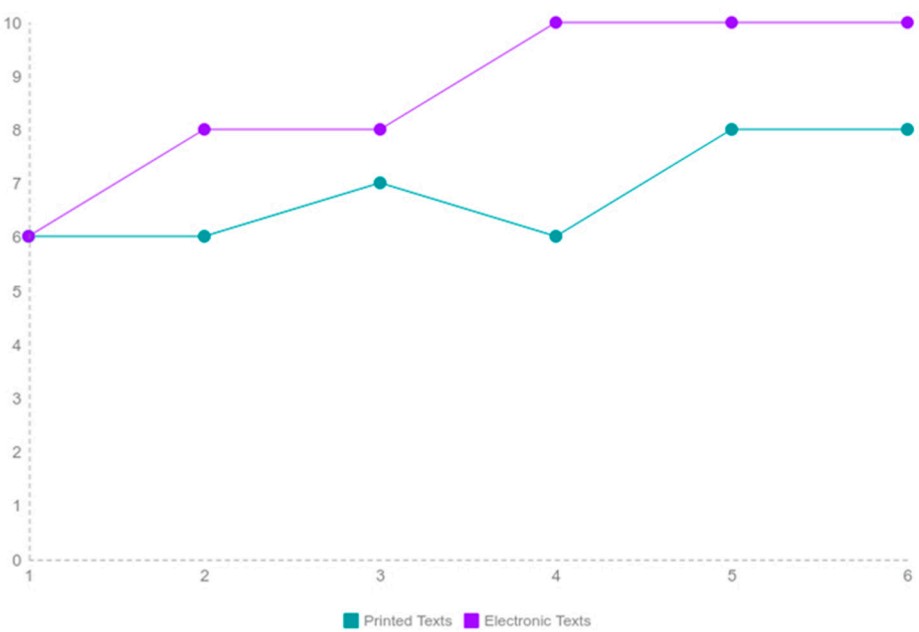

**Figure 5.** Literal skills of Hikmet.

**Table 7.** Comparison of the efficiency of printed texts and electronic texts in terms of word acquisition skills.

| Name of the Subject | Type of Text | Total Number of Sessions | Number of the Session When the Criterion Was Met | Total Number of True Answers | Total Number of False Answers |
|---|---|---|---|---|---|
| Hikmet | Printed: 6 Electronic: 6 | 12 | Printed: 2 Electronic: 5 | Printed: 15 Electronic: 26 | Printed: 15 Electronic: 4 |
| Arda | Printed: 6 Electronic: 6 | 12 | Printed: 0 Electronic: 5 | Printed: 16 Electronic: 26 | Printed: 14 Electronic: 4 |

**Table 8.** Comparison of the efficiency of printed texts and electronic texts in terms of literal skills.

| Name of the Subject | Type of Text | Total Number of Sessions | Number of the Session When the Criterion Was Met | Total Number of True Answers | Total Number of False Answers |
|---|---|---|---|---|---|
| Hikmet | Printed: 6 Electronic: 6 | 12 | Printed: 0 Electronic: 4 | Printed: 40 Electronic: 54 | Printed: 20 Electronic: 6 |
| Arda | Printed: 6 Electronic: 6 | 12 | Printed: 0 Electronic: 3 | Printed: 36 Electronic: 52 | Printed: 24 Electronic: 8 |

*3.3. Comparison of Efficiency of Printed Texts and Electronic Texts in Terms of Vocabulary Acquisition Achievement Levels*

The number of sessions until the criterion was met between the printed and electronic texts presented to the subjects in the evaluation of vocabulary acquisition achievement levels was tabulated by calculating and comparing the number of true and false answers (Table 7).

Calculations were made to evaluate whether there were any differences between the printed texts and the electronic texts in terms of the efficiency of vocabulary acquisition.

According to Table 7, in which the efficiency of printed texts and electronic texts are compared in terms of vocabulary acquisition levels, in the study conducted using 12 texts including six printed and six electronic texts, it is observed that the first subject, Hikmet, met the criterion determined with printed texts a total of two times, and the criterion determined with electronic texts a total of five times. The second subject, Arda, was never able to fulfill the criterion determined by printed texts (0), but he met the criterion for electronic texts in a total of five sessions. In addition, when the total number of true and false answers are compared according to both subjects and the types of texts, it can be seen that the first subject, Hikmet, gave 15 true and 15 false answers in total to the printed texts, whereas he gave 26 true answers and 4 false answers to the electronic texts. In regard to the second subject, Arda, it is observed that he gave 16 true and 14 false answers in total to the printed texts, whereas the number of true and false answers he gave to the electronic texts was also 26 and 4, respectively.

Based on the above findings, it can be seen that both subjects gave true and false answers in similar proportions to the printed and electronic texts, and the true answers to the questions asked in electronic texts were as almost double those related to the printed texts. This is believed to be due to the fact that electronic texts provide a more sensual and unusual presentation, which is of interest to children. It is seen that the students provided more false answers when they were given printed (plain) texts, whereas this situation changed in electronic texts, and the number of true answers increased. Considering that the students met the criteria related to vocabulary acquisition levels five times in the sessions conducted through the electronic books, and obtained results ranging from 0 to 2 in the printed texts, it is believed that electronic texts had an impact on the vocabulary acquisition levels of the students.

*3.4. Comparison of the Efficiency of Printed Texts and Electronic Texts in Terms of Literal Skills*

In terms of the evaluation of the development of literal skills, the number of sessions realized until the criterion were met between the printed and electronic texts presented to the subjects has been tabulated based on the calculation and comparison of the number of true and false answers (Table 8), and it has been determined whether there are any differences between the printed texts and the electronic texts in terms of efficiency in the development of literal skills.

When Table 8, which compares the efficiency of printed and electronic texts in terms of literal skills, is examined (in the study conducted using 12 texts in total, six printed and six electronic texts), it is seen that the first subject, Hikmet, did not meet the criteria for literal skills with printed texts (0), but he met the criteria determined for electronic texts a total of four times. It can be seen that Arda, the second subject, never fulfilled the criterion for the assessment of literal skills with printed texts (0), but he was able to meet the criteria in three sessions with electronic texts. In addition, when a

comparison between the total true and false answers is made according to the subjects and type of texts, it can be seen that the first subject, Hikmet, gave 40 true and 20 false answers in the printed texts, whereas he gave a total of 54 true and 6 false answers in electronic texts. It is seen that the second subject, Arda, gave 36 true and 24 false answers in printed texts, whereas he gave 52 true and 8 false answers in electronic texts in total.

Based upon the above findings, it can be argued that although the subjects' literal skills improved using both text types, better results were obtained from applications with electronic texts. Therefore, it can be argued that electronic texts were more effective on the literal skills of students.

### 3.5. Evaluation of Social Validity Results Obtained from Students

When the data in Table 9, (which compares the printed and electronic texts according to the social validity results of the 1st subject, Hikmet), are examined, it is understood that Hikmet found reading from a computer (electronic texts) more fun than reading from a book (printed texts) in accordance with the answers he gave to the questions. He felt successful and happy when reading from a computer (electronic text). Although he stated that he lost his place while reading both from a book (printed texts) and a computer (electronic texts), he also mentioned that he could easily understand what he read on the computer (with electronic texts), but could not easily understand what he read in the book (printed texts). Hikmet also stated that he could understand the meaning of new words easily when reading from a computer (electronic texts), whereas he could not understand the meaning of new words easily when reading from book (printed texts). When the above findings are examined, it is considered that the reason why Hikmet felt happier, more successful and understood what he read on electronic texts, is that it was more interesting for him to read from a screen instead of the traditional method of reading from printed text.

**Table 9.** Comparison of printed and electronic texts according to the social validity results of the two subjects.

| Social Validity Questions | Hikmet's Answers | Arda's Answers |
|---|---|---|
| I think that reading from a book/computer is more enjoyable. | From computer | From computer |
| When reading from a computer, I feel....... | Successful and happy | Amused and cheerful |
| When reading from a book, I feel...... | Bored | Concerned and sometimes unsuccessful |
| I can lose my place when reading from a computer. (Yes/No) | Yes | Yes |
| I can lose my place when reading from a book. (Yes/No) | Yes | No (I trace with my finger) |
| When reading on a computer, I can easily understand what I read. (Yes/No) | Yes | Yes |
| When reading from a book, I can easily understand what I read. (Yes/No) | No | No |
| I can easily comprehend the meaning of new vocabulary when reading from computer. | Yes | Yes |
| I can easily comprehend the meaning of new vocabulary when reading from a book. | No | No |

When the social validity results of the 2nd subject, Arda, are examined, it can be observed, based on the answers which he provided to the questions, that he found it more entertaining to read from a computer (electronic texts) than to read from books (printed texts). Arda also stated that he could easily understand the texts when he read on the computer (electronic texts), whereas he could not easily comprehend when reading from books (printed texts).

Arda stated that he could easily understand the meaning of the new words when reading from the computer (electronic texts), but he could not understand the meaning of the new words all that easily when reading from the book (printed texts). When the above findings are analyzed, it is believed that the reason why Arda felt happier, more entertained and joyful when reading electronic texts, is that electronic texts aroused his interest in the subject.

## 4. Discussion and Conclusions

The objective of this study is to evaluate the vocabulary acquisition achievement level and literal skills of students with reading comprehension difficulties by presenting printed (plain) text and electronic texts to them, and to determine whether there are any differences between the two types of texts in terms of the development of these skills. According to the results of the study, the vocabulary acquisition levels of both students gave more effective results in presentations made with electronic texts. In other words, the students' level of word acquisition was found to be more effective than the results obtained after the presentation with printed texts. In this respect, the results of the study are in line with the findings of other studies conducted in the literature [36,42,46,47]. According to these researchers, when compared to printed books, it is found that electronically designed books contain multimedia effects such as written text, oral reading, oral discourse, music, sound effects and animations, and the verbal reading of the text by the narrator accompanied by the highlighted text, which also helps children to trace the written words, phrases or passages being read carefully, and helps them to read and acquire vocabulary [36,46,47,53,54].

According to the results obtained from the examination of the literal skills by comparing the printed (plain) presentations with the texts presented with electronic presentations, the response rate of the students to the questions presented in the electronic media was higher than the printed texts. In other words, electronic texts were found to be more effective on improving literal skills than printed texts. When compared with the results of other studies conducted in the literature, it is seen that the results obtained in this study are in line with others reported in the literature [55–58]. According to the research findings, the participants had more self-confidence and willingness to participate, a higher number of true answers were given to the questions about the texts read, and the willingness to read was higher in the sessions conducted with electronic texts in comparison to those with printed texts.

While the findings obtained from the research generally show that electronic texts provide more effective results on the levels of vocabulary acquisition and the literal skills levels of the participants, other studies in the literature have reported opposite results. Accordingly, some studies [59–63] have concluded that printed materials have a greater effect on participants' literal skills than electronic texts. In the literature, there are also studies reflecting that there is no difference in terms of the impact between printed and electronic texts [62,64–67].

In this study, it was observed that electronic texts increased the number of correct answers given by the students, and positively affected their desire to read. However, this should not mean that printed texts should be completely disregarded. It can be said that printed texts may have some advantages, and electronic texts may have some disadvantages. For example, the advantages of printed texts in the reading process include the ability to take notes and highlight important points [68]. On the other hand, the disadvantages of electronic texts are the inability to see the whole page, early eye fatigue, and a difficulty in returning back and read again. Through an awareness of the advantages and disadvantages of electronic and printed texts, efforts should be to minimize the problems that can be experienced during the teaching process [69].

According to the social validity data, the students in this study stated that they had never had the opportunity to experience electronic texts during their education lives. In other words, the use of electronic texts represented a relatively new phenomenon for them. If used frequently and after a certain period of time, it is likely that their interest in these texts will diminish as they become accustomed to electronic texts. As Grimshaw, Dungworth, McKnight and Morris stated [54], it is

necessary to apply different activities and evaluation techniques in the process of using electronic texts in order to prevent students' interest from being decreased.

Studies where students record their own voice while reading the text, write the answers to the questions on the computer, use a digital dictionary, and research and find visuals related to the text on the internet, can be performed.

Finally, it is believed that, different from this study, it is important to relate the e-reading experiences of students with special needs and those with normal development in further studies. In some studies in the literature [70,71], it has been emphasized that using assistive technologies provides effective results for special education students, as revealed in this study. E-reading experiences and the success of children with special needs can be evaluated, and positive results can be used. For example, success stories in special education studies can also be a guide for the education of students who develop normally in inclusive classroom settings. In addition, a contribution can be made to enhance the effectiveness of the teaching skills of teachers and their ability to perform teaching adaptations.

## 5. Limitations and Recommendations

When the findings of the study are evaluated, some limitations can be observed. All of the texts used in the experimental practice were narrative. This can be seen as a limitation in the study, since it was not tested how the informative type of texts would affect the results of the study. On the other hand, reading fluency, which is directly influenced by literal skills according to the literature, has not been evaluated in this study. This can be considered as another limitation. The final limitation of the study is that all questions posed in the experimental practice consisted of items that measured literal skills. Therefore, no data could be obtained in regard to whether inferential skills were developed, because only one aspect of reading comprehension was evaluated.

The recommendations related to the research results are presented below:

1. Using assistive technologies such as computers, tablets and overhead projectors in the classroom will be useful in increasing the effectiveness and permanence of learning activities, especially where both students with special education needs, and also their peers, exist.
2. To increase the number of true answers given by the students in the printed texts, it is suggested that teachers perform additional studies; in other words, they could make teaching adaptations and utilize reading comprehension strategies (strategies used prior to, during and after reading). Particularly considering that the students' interest in electronic texts was not observed in written texts, it may be useful to carry out additional studies on this subject (e.g., making the texts interesting by supporting them with visual materials).
3. The number of applied studies in the field of special education that examine the effectiveness of printed and electronic texts can be increased.
4. Similar studies can also be carried out with students in other special needs groups in other courses.

**Author Contributions:** Conceptualization, Ö.D.G.; methodology, Ö.D.G.; validation, Ö.D.G.; investigation, Ö.D.G.; resources, Ö.D.G.; data curation, Ö.D.G.; writing—original draft preparation, Ö.D.G.; writing—review and editing, A.G.; supervision, A.G.; project administration, A.G.

**Funding:** This research received no external funding.

**Conflicts of Interest:** The authors declare no conflict of interest.

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
