# Peer review of "Printed Versus Electronic Texts in Inclusive Environments: Comparison Research on the Reading Comprehension Skills and Vocabulary Acquisition of Special Needs Students"

_education, doi:10.3390/educsci9030246_

Round 1

Reviewer 1 Report

I appreciated the opportunity to review the submitted manuscript regarding the different reading materials and outcomes in comprehension and vocabulary for students with disabilities. There are several major issues that need to be addressed within the manuscript for it to be publishable. 

Comprehension is a large concept with many components regarding how text is processed by an person. The broad use of the term comprehension needs to be refined within the paper. It appears that the measure is only of "wh" questions- it is not specified if these are literal or inferential but given the age of the participants it is assumed literal, but that should be spelled out. Clarifying that this paper only measured that one aspect of comprehension is necessary. 

The introduction and various other sections are very difficult to follow in content and structure. Reading it for clarity (e.g., it is unclear what point you are trying to make in lines 31-34; lines 41-41- what is the difference between a negative attitude towards reading and the inability to enjoy reading?; I am unclear what is the point the author is summarizing in lines 55-57, lines 69-70; what is "form characteristics" in line 77, line 212 should be "for" instead of "against", the entire paragraph beginning line 270, etc.) and format (e.g., a single sentence should not be a paragraph; the sentence structure is not uniform in lines 50-53). The entire introduction is riddled with clarity issues. 

The information regarding the participants should include the type of disability for each student. Also the fourth inclusion criteria and the fifth column in the table about the participants seems to be in disagreement. 

Is "productivity" the correct term for the comparison the authors made for research question 4? It seems that your table is about efficiency, not productivity. 

Was the schedule of Table 2 the same schedule for both students? It would have been a stronger study for validity had you alternated the story schedule across the two students to be sure the effect wasn't text and time of day dependent. Did you do anything to ensure the difficulty of the text was equated across texts? If so that should clearly be reported with formal data.

It would be good to include samples of the questions and include a table of all of the vocabulary from each story. 

The measure or checklist should clearly be described when you explain the dependent variable starting line 234. The paragraph starting line 270 is confusing as written about practices and further assessments being made on correct responses. What does that mean? What practices is the author referring to? 

Graphs are described in the analysis of the data but no graphs are included in the manuscript. Including the graphs will help increase the confidence in the reported data. 

The IOR data is incomplete- line 305.

Somewhere in the manuscript, it should be clarified that the maximum correct score for vocabulary for each story was 5 and was 10 for questions. It would be helpful in the interpretation of the tables to have that information included near each table. 

Without any data about the difficulty of stories or of vocabulary words, it is difficult to ensure that the interpretations of the results are accurate as written. 

The authors only include descriptive data. "Significance" cannot be established with data from two participants in the way this data is reported. All references to significance (e.g., line 315, 362,  398, 428, 489, etc.) should be removed. 

Be careful linking your study to other studies that used narrated electronic text- those studies measured listening comprehension- a very different skill than you measured. 

The recommendation section appears incomplete and is not formatted correctly for a manuscript. 

Author Response

Reviewer 1

Point 1. Comprehension is a large concept with many components regarding how text is processed by an person. The broad use of the term comprehension needs to be refined within the paper. It appears that the measure is only of "wh" questions- it is not specified if these are literal or inferential but given the age of the participants it is assumed literal, but that should be spelled out. Clarifying that this paper only measured that one aspect of comprehension is necessary.

Instead of “comprehension” concept, the “literal skills” concept is used in the article. Please check line 16, line 40 etc

Point 2. The introduction and various other sections are very difficult to follow in content and structure. Reading it for clarity (e.g., it is unclear what point you are trying to make in lines 31-34; lines 41-41- what is the difference between a negative attitude towards reading and the inability to enjoy reading?; I am unclear what is the point the author is summarizing in lines 55-57, lines 6970; what is "form characteristics" in line 77, line 212 should be "for" instead of "against", the entire paragraph beginning line 270, etc.) and format (e.g., a single sentence should not be a paragraph; the sentence structure is not uniform in lines 50-53). The entire introduction is riddled with clarity issues.

Please check the letter of proofreading written by a native English speaker. The article is read by Proofreader & Editor.

Point 3. The information regarding the participants should include the type of disability for each student. Also the fourth inclusion criteria and the fifth column in the table about the participants seems to be in disagreement.

An information paragraph is added from line 125 to line 133.

Point 4. Is "productivity" the correct term for the comparison the authors made for research question 4? It seems that your table is about efficiency, not productivity.

Research questions are changed and question 4 is deleted.

Point 5. Was the schedule of Table 2 the same schedule for both students? It would have been a stronger study for validity had you alternated the story schedule across the two students to be sure the effect wasn't text and time of day dependent. Did you do anything to ensure the difficulty of the text was equated across texts? If so that should clearly be reported with formal data. Without any data about the difficulty of stories or of vocabulary words, it is difficult to ensure that the interpretations of the results are accurate as written.

For this comment, researchers added some explanations from line 205 to line 219 

Point 6. It would be good to include samples of the questions and include a table of all of the vocabulary from each story.

Attachment 1 is added.

Point 7. The measure or checklist should clearly be described when you explain the dependent variable starting line 234. The paragraph starting line 270 is confusing as written about practices and further assessments being made on correct responses. What does that mean? What practices is the author referring to?

Checklist is described in the section of “2.5.1. Dependent and Independent Variables” from line 261 to line 269.

Point 8. Graphs are described in the analysis of the data but no graphs are included in the manuscript. Including the graphs will help increase the confidence in the reported data.

Graphs are described in line 408, line 419, line 469, line 471.

Point 9. The IOR data is incomplete- line 305.

The IOR data is completed in line 344

Point 10. Somewhere in the manuscript, it should be clarified that the maximum correct score for vocabulary for each story was 5 and was 10 for questions. It would be helpful in the interpretation of the tables to have that information included near each table.

The explanation is added for the maximum correct score for vocabulary (which is 5) and literal skill questions (which is 10). Please check line 406 and line 454.

Point 11. The authors only include descriptive data. "Significance" cannot be established with data from two participants in the way this data is reported. All references to significance (e.g., line 315, 362,  398, 428, 489, etc.) should be removed.

The concept of “significance” was removed.

Point 12. Be careful linking your study to other studies that used narrated electronic text- those studies measured listening comprehension- a very different skill than you measured.

All of the references were checked again and there weren’t any results that are related with listening comprehension.

Point 13. The recommendation section appears incomplete and is not formatted correctly for a manuscript.

The recommendation section is rewritten by researchers. Please check line 617 to 638.

Reviewer 2 Report

Thank you for submitting your manuscript.

I have reviewed your paper, and regret to inform you that there are important difficulties in it.

The manuscript has the potential to make a contribution to the literatura. I have significant concerns about the manuscript as witten:

The writing is difficult to follow in many places, with grammatical and English language usage errors obscuring meaning. The goals of the study are a bit obscure and not well motivated. For example, how to talk about a significant difference with two subjects? More information needs to be provided about the methods, type of analysis or observational methodology. The characteristics of the only two subjects in the study are unknown. Being an in-depth study, it is necessary to detail their special educational needs, as well as other sociodemographic and personal children's characteristics. The results are presented in a simple way and the data related to the inter-observer agreement are not detailed. The discussion is quite superficial, mostly restating the findings in short one- or two-sentence paragraphs. More interpretation and integration is needed to make a significant contribution. They had to add a section of limitations of the study because it is such a limited sample The title should have been more indicative of the limited study carried out.

I appreciate the opportunity to consider your work.

Author Response

Point 1. The writing is difficult to follow in many places, with grammatical and English language usage errors obscuring meaning.

Please check the letter of proofreading written by a native English speaker. The article is read by Proofreader & Editor.

Point 2. The goals of the study are a bit obscure and not well motivated. For example, how to talk about a significant difference with two subjects?

Goals of the study were changed. The concept of “significant difference” is deleted.

Point 3. More information needs to be provided about the methods, type of analysis or observational methodology.

More information is added to Methods. Please check line 125-133, 205-219, 261-269, 307-313, 338-342.

Point 4. The characteristics of the only two subjects in the study are unknown. Being an in-depth study, it is necessary to detail their special educational needs, as well as other sociodemographic and personal children's characteristics.

An information paragraph about subjects is added from line 125 to line 133.

Point 5. The results are presented in a simple way and the data related to the interobserver agreement are not detailed.

Interobserver agreement is detailed in line 338 to 342

Point 6. The discussion is quite superficial, mostly restating the findings in short one- or two-sentence paragraphs. More interpretation and integration is needed to make a significant contribution.

3 more paragraphs were added (from line 589 to 615) to discussion to make a significant contribution

Point 7. They had to add a section of limitations of the study because it is such a limited sample.

The recommendation section is written by researchers. Please check line 617 to 625.

Point 8. The title should have been more indicative of the limited study carried out.

The title is changed.

Round 2

Reviewer 1 Report

The authors addressed most of the points provided in the first review.

The replacement of "comprehension"with "literal skills" added to the confusion. The Introduction should continue to use "comprehension" but the method section should describe that this study only addressed answering wh questions as one aspect of comprehension. The clarity of what is being measured- wh questions- should be explained in the tools and dependent variable sections.  One previous review point about the conflicting information about criteria for inclusion between the Table 1 (fifth column) and the fourth criteria point remains.  Person-first language (students with special needs) should be included throughout the paper.  A solid read once all the track changes are accepted should be conducted to ensure grammar use. For example, in the title, the word "A" should be removed after the colon. Be sure to avoid single sentence paragraphs. Check your replacements for the use of "Assistive technology" versus "Supporting technology". A reference is needed after "inclusive education environments" on line 41.  The additional paragraph about subjects isn't needed as written. Identifying the students as having a learning disability with difficulties in reading comprehension was the only addition needed as requested. It is suggested to only add that information and delete the remaining information in the newly added paragraph.  Don't identify the school affiliation if it isn't needed in line 140. By not including the information the participants are less likely to be identifable.  

Author Response

Dear Reviewer,

Below, you can find my comments according to your response.

Point 1. The replacement of "comprehension" with "literal skills" added to the confusion. The Introduction should continue to use "comprehension" but the method section should describe that this study only addressed answering wh questions as one aspect of comprehension. The clarity of what is being measured- wh questions- should be explained in the tools and dependent variable sections.

We changed “literal skills” as “comprehension” in introduction part. The clarity of what is being measured is explained in tools and dependent variable sections. Please check line 228-231 and 258-259.

Point 2. One previous review point about the conflicting information about criteria for inclusion between the Table 1 (fifth column) and the fourth criteria point remains.

Fourth criteria in Table 1 is changed like “number of correct answers given to vocabulary questions”

Point 3. Person-first language (students with special needs) should be included throughout the paper.

First language of both students with special needs is added. Please check line 141-142

Point 4. A solid read once all the track changes are accepted should be conducted to ensure grammar use. For example, in the title, the word "A" should be removed after the colon.

Second proofreading is done by native English teacher. Please see the supplementary file.

Point 5. Be sure to avoid single sentence paragraphs.

There isn’t any single sentence paragraphs

Point 6. Check your replacements for the use of "Assistive technology" versus "Supporting technology".

“Supporting technology” is changed and assistive technology is used in the article.

Point 7. A reference is needed after "inclusive education environments" on line 41. 

A reference is added after inclusive educational environments. Please check line 43

Point 8. The additional paragraph about subjects isn't needed as written. Identifying the students as having a learning disability with difficulties in reading comprehension was the only addition needed as requested. It is suggested to only add that information and delete the remaining information in the newly added paragraph. 

The detailed information about subjects is deleted and the only addition is about reading comprehension disability. Please check  line129-131.

Point 9. Don't identify the school affiliation if it isn't needed in line 140. By not including the information the participants are less likely to be identifable. 

We didn’t understand this suggestion because there isn’t any information about school affiliation. Please check line 139-141.

Reviewer 2 Report

The paper has improved and can be published.
Best regards.

Author Response

Dear Reviewer,

I would like to thank you four your important comments to improve my articles quality.

Kind regards.